# Dispositional Mindfulness and Inhibitory Control after Perceiving Neutral, Food and Money Related Stimuli

**DOI:** 10.3390/ijerph19042201

**Published:** 2022-02-15

**Authors:** Zsófia Logemann-Molnár, Atakan M. Akıl, Renata Cserjési, Tamás Nagy, Anna Veres-Székely, Zsolt Demetrovics, Alexander Logemann

**Affiliations:** 1Doctoral School of Psychology, ELTE Eötvös Loránd University, 1064 Budapest, Hungary; logemann.zsofia@ppk.elte.hu (Z.L.-M.); atakan.akil@ppk.elte.hu (A.M.A.); 2Institute of Psychology, ELTE, Eötvös Loránd University, 1064 Budapest, Hungary; cserjesi.renata@ppk.elte.hu (R.C.); nagy.tamas@ppk.elte.hu (T.N.); demetrovics.zsolt@ppk.elte.hu (Z.D.); 3MTA-ELTE Lendület Adaptation Research Group, Institute of Psychology, ELTE Eötvös Loránd University, 1064 Budapest, Hungary; szekely.anna@ppk.elte.hu; 4Centre of Excellence in Responsible Gaming, University of Gibraltar, Gibraltar GX11 1AA, UK

**Keywords:** mindfulness, inhibition, stop signal task, SSRT, reward, money

## Abstract

Previous studies have shown that dispositional mindfulness is positively associated with cognitive performance, including the ability to stop behavioral actions (formally called inhibitory control). However, some studies suggest that the relationship may be context dependent. The current study addressed previous limitations and focused on the role of reward context regarding the relationship between mindfulness and inhibitory control. Seventy-five participants (31 men, 44 women) between 18–50 years old (M = 30, SD = 9) were included in the final sample. Participants filled out a self-report measure of mindfulness and performed a stop signal task with three conditions that varied in terms of reward context. In the neutral condition, go stimuli (stimuli to which a response was required) were letters; in the food and money condition, these were pictures of food and money, respectively. Results showed that inhibitory control was reduced in the money condition relative to the neutral condition (*p* = 0.012). Mindfulness was positively correlated with inhibitory control, but only in the money condition (*p* = 0.001). However, results might differ when inhibitory control is required while perceiving a learned reward-related stimulus. The latter represents an open question for future research.

## 1. Introduction

Inhibitory control is important for adaptive everyday functioning. In fact, problems of inhibitory control can have significant consequences, and deficits have been implicated in psychopathology such as Attention-Deficit/Hyperactivity Disorder [1], drug addiction [2], and obesity [3,4]. Previous studies suggest that mindfulness training is positively associated with executive functions, including inhibitory performance [5,6], which obviously has clinical implications. Less is known about dispositional mindfulness in relation to inhibitory control, though previous studies have indicated that high dispositional, or in other words trait, mindfulness is associated with improved executive functioning using mixed measures [7,8].

Mindfulness can be described as the capacity of individuals to be aware of and attend both internal and external events in a non-judgmental, open, and discerning way [9]. Previous studies have suggested that mindfulness is associated with improved cognitive performance [6,10,11]. Specifically, mindfulness training has been shown to enhance go/no-go type task performance [5], which may suggest an inhibitory improvement. In such a task, visual go stimuli to which a simple response is required are presented sequentially. Infrequently, a visual no-go stimulus is presented to which a response has to be withheld. Importantly, it should be noted that the relevant behavioral outcome may not only be affected by inhibitory processes but also by other processes (such as processing speed and attention). Specifically, speeded response tendencies due to attentional bias logically negatively affect inhibitory performance on such a task.

Complicating the matter further is that the exact implementation of the go/no-go task varies wildly in the literature, and whether inhibitory control is measured at all depends strongly on specific task parameters (most notably no-go stimuli onset frequency and intertrial duration) [12]. Perhaps not surprisingly, results regarding the relationship between mindfulness and behavioral reflections of inhibitory control vary. With some reporting a significant relationship [5], and others that do not find clear support for such relationship [13,14]. Taken together, even though previous studies suggest a positive relationship between mindfulness and inhibitory control, the relationship might be explained, at least in part, by other processes.

To the best of our knowledge, the exact mechanism that explains a potential relationship between mindfulness and inhibitory control is yet to be elucidated, though it has been suggested that a mindful state is associated with enhanced top-down executive control over behavior and attenuated bottom-up driven (reflexive) processes [15,16]. This is important, as relevant environmental cues such as stimuli associated with rewards may reflexively capture attention [17] and may challenge subsequent inhibitory control, especially in addiction [18]. In fact, several studies have reported that exposure to reward-related stimuli may significantly challenge inhibitory control, especially in addiction [19,20] and obesity [4,20]. Hence, if mindfulness is associated with improved inhibitory control and reduced impact of bottom-up driven processes, the positive effect of mindfulness on inhibitory control may be especially strong in reward contexts.

However, recent studies seem to suggest an inverse relationship. For instance, a recent study investigated the effect of a brief mindfulness intervention on smokers’ performance and electrophysiological correlates in a smoking go/no-go task [13]. In their go/no-go task, neutral and smoking-related stimuli were randomly and sequentially presented, to which, in most cases, a response was required (go-stimuli). In infrequent no-go trials, a response had to be withheld subsequent to a no-go stimulus. Whether the stimulus was a go-stimulus or no-go stimulus was determined by the border color. Though no group effect was found on the behavioral index of inhibitory performance (proportion of inhibitions to no-go stimuli), the no-go P3 event-related potential was found to be smaller in the mindfulness group relative to the control group. As the no-go (or stop), P3 is thought to reflect inhibitory control, a smaller P3 can be taken as reflecting reduced inhibitory control, in this case in a reward context subsequent to mindfulness intervention [12,21]. Recent results from our previous project are congruent with these findings [22]. In that aforementioned study, participants filled out a questionnaire measuring dispositional mindfulness and performed an adapted go/no-go (GNG) task with a neutral and social-reward context. It was shown that higher mindfulness was associated with a reduced proportion of successful inhibitions on no-go trials in the social reward context. Though it is not yet clear what explains the negative relationship between mindfulness and inhibitory control in a reward context, it seems logical that being aware of internal and external events in the present (one main aspect of mindfulness) results in increased susceptibility to relevant environmental stimuli, such as stimuli indicating reward.

One key limitation of previous studies is the use of paradigms that yield an outcome measure that reflects processes other than, or in addition to, inhibitory control. In the current study, we addressed this limitation to answer the aforementioned questions. To assess the relationship between dispositional mindfulness on a pure measure of inhibitory control, we employed an online adapted stop signal task (SST). The SST seems somewhat similar to a standard go/no-go task; however, in the SST, a go stimulus is infrequently followed by a stop signal, requiring the prepotent response to be withheld. In contrast to the main outcome variable in the GNG task, the SSRT is not, or at least less, sensitive to variations in response speed due to, e.g., response bias. To address the question regarding the potential moderating role of reward context, our SST included three conditions that differed in terms of reward context. In the neutral condition, go stimuli were letters. In the reward conditions, stimuli were related to generally recognizable palatable foods, foods that are high in fat and sugar (food condition), or money-related (money condition). We chose these two different types of reward contexts to evaluate whether the relationship between mindfulness on inhibitory control, when exposed to one type of reward, would be mirrored in the other. That would be expected, as both learned reward-related stimuli (such as money) and intrinsic reward-related stimuli (such as palatable food) are known to trigger main brain regions responsible for reward processing [23].

Taken together, the main aim of the current study was to address the limitations of previous studies and assess the relationship between dispositional mindfulness and inhibitory control in conditions that differed in terms of reward context. Based on the above-discussed literature, we hypothesized that there would be a positive relationship between mindfulness and inhibitory control in the neutral condition (as reflected by a negative correlation between trait mindfulness score and stop signal reaction time). Secondly, we hypothesized that higher mindfulness would be associated with challenged inhibitory control in both reward conditions, reflected by a positive correlation between trait mindfulness and stop signal reaction time in these conditions.

## 2. Methods

### 2.1. Participants

Participants were recruited via social media and via Prolific (Prolific. Available online: http://www.prolific.co (accessed on 20 January 2021)). Individuals could participate if they were 18–50 years old, did not suffer from any psychological or neurological disorder, and did not use drugs within seven days prior to the experiment. We aimed to detect moderate correlations. The target sample size was estimated with G*Power [24]. Specifically, with alpha set at 0.05 and power at 0.8, the initial target sample size was 80 participants. In total, 87 participants completed the experiments and were provided with a 2.50 GBP monetary compensation for participating. In total, 64 participants were from the United Kingdom and were native English speakers. The other 23 participants were from elsewhere and not necessarily native English speakers. However, participants were excluded from the subsequent analyses if they did not meet at least a B1 independent level of lingual proficiency in English (according to the Language proficiency questionnaire). In addition, we excluded participants with an invalid estimate of the SSRT (see the materials section for details). The final sample consisted of 75 participants (31 men, 44 women) between 18–50 years old (M = 30, SD = 9). This sample size was sufficient to detect moderate effect sizes of r > 0.32. All participants signed the informed consent form prior to any experimental procedures. The experiment was approved by the Research Ethics Committee of the Institute of Psychology, Eötvös Loránd University (ELTE), and conducted in accordance with the Declaration of Helsinki.

### 2.2. Materials

#### 2.2.1. Psytoolkit

All behavioral and self-report assessments were implemented in Psytoolkit, a platform for developing and running online cognitive psychological experiments [25,26]. Previous research indicates strong replicability, even of a complex experiment requiring short-latency response logging, comparing Psytoolkit to an offline implementation using high-end equipment and software [27] in a controlled environment [28].

#### 2.2.2. Mindful Attention Awareness Scale

Dispositional mindfulness was assessed with the 15 item MAAS [29]. One example item is, for example, “I could be experiencing some emotion and not be conscious of it until some time later”. Participants answer the items on a six-point scale ranging from “Almost always (score 1)” to “Almost never (score 6)”. The MAAS score is the average item score. A higher score represents a higher level of mindfulness. The scale reliability is known to be high with a Cronbach’s alpha exceeding 0.8. In our final sample (*n* = 75), Cronbach’s alpha was 0.86.

#### 2.2.3. The Stop Signal Task

The implemented online stop signal task (SST) was based on its original conceptualization [30] and was adapted based on a previous implementation [4]. Go stimuli required a response, either a left-button press or a right-button press with the index finger. Go stimuli were infrequently followed by a stop stimulus, the letter “S” (115(w) × 200(h) pixels). The stop signal was required to withhold the prepotent response. Go and stop stimuli were presented centrally, slightly above a persistent central fixation dot. Stimulus duration was fixed at 150 ms. The intertrial interval was 1500 ms for go trials, and 1700 ms for stop trials. Initially, the interval between the onset of the go stimulus and stop stimulus was set at 250 ms, then dynamically adjusted after each stop trial via tracking algorithm. Specifically, in case of a failed inhibition, the go-stop interval was decreased by 50 ms. In case of a successful inhibition, the go-stop interval was increased by 50 ms. The tracking algorithm yields an approximate 50% inhibit rate, increasing the reliability of the estimation of the SSRT [31]. The task included a practice condition and the experimental conditions; the neutral, food, and money condition. The practice block consisted of 10 go trials and 10 stop trials. In the practice block, participants were provided with immediate feedback when they made an error of omission, commission, and/or made an incorrect response to the go stimulus. The experimental conditions each consisted of 48 go trials and 16 stop trials. The experimental conditions differed only in terms of the go-stimuli. In the neutral condition, go stimuli were two equiprobable letters: X and O (both 150(w) × 195(h) pixels) randomly presented, and the required response depended on the letter presented. Similar to a previous implementation [4], in the food condition, the go stimulus was one of four equiprobable pictures of palatable food (chips, chocolate-chip cookies, nuts, chocolate), presented in either landscape or portrait orientation. In the money condition, the go stimulus was one of four equiprobable pictures of money most individuals are familiar with (Euros and Dollars), also presented in either landscape or portrait orientation. In these latter two conditions, the response depended on the orientation of the stimulus. Trials were randomized for each participant, and the condition order of the experimental conditions was counterbalanced across participants. The SSRT was calculated using the integration method [31], and inhibitions were corrected for omissions [32]. To elaborate, the corrected proportion of inhibitions was the uncorrected proportion of inhibition—(proportion of omissions × proportion of inhibitions)/(1 − (proportion of omissions × proportion of inhibitions)). Response times (RTs) shorter than 150 ms and exceeding the intertrial interval were discarded. RTs in go trials were rank-ordered from short to long RT, and the nth point on the rank-ordered distribution of RT scores was determined by multiplying the length of the RT vector by one minus the corrected inhibit rate. Finally, SSRT was estimated by subtracting the mean go-stop stimulus asynchrony from the RT value at the nth point on the rank-ordered RT distribution. We applied an a priori defined lenient exclusion criterion to the SST data, which adds to a reliable estimation of SSRT [33].

#### 2.2.4. Language Proficiency According to the Common European Framework of Reference for Languages-Self-Assessment Grid

The language proficiency questionnaire included one question, “Please indicate your English language reading proficiency”, with six response options reflecting the level of proficiency according to the Common European Framework of Reference for Languages. These levels are “A1 Basic user”, “A2 Basic user”, “B1 Independent user”, “B2 Independent user”, “C1 Proficient user”, and lastly, “C2 proficient user”. One exemplar statement reflecting the “B1 Independent user” level of proficiency was: “I can understand texts that consist mainly of high-frequency everyday or job-related language. I can understand the description of events, feelings, and wishes in personal letters.” The latter represented the minimum level of English language proficiency required in the current research.

### 2.3. Procedure

Participants were recruited via social media and prolific.co and were provided with a 2.50 GBP monetary incentive. Subsequent to reading the information letter and signing the informed consent form, participants started with filling out the questionnaires. Subsequently, participants were provided with the instruction of the stop-signal task. Specifically, they were informed that the goal was to respond as quickly and accurately as possible to the go stimulus and to withhold response in case the stop signal was presented. To aid understanding of the instruction, participants were provided with feedback during the practice part of the task. After participants performed all conditions, the experiment was completed.

### 2.4. Data Analysis

The outcome variables were calculated using R [34], and subsequent statistical analyses were performed with SPSS [35]. Analyses were planned apriori with alpha at 0.05. Specifically, to assess the overall main effect of the condition, we planned an ANOVA with condition (neutral, food, or money) as independent variable and SSRT as the dependent variable. For the relationship between MAAS score and SSRT in the different conditions, Pearson’s test of correlation was planned for each condition. Prior to any analyses, assumptions were verified.

## 3. Results

Performance data are summarized in Table 1. Following the exclusion criteria as outlined in a previous report [33], we excluded participants with negative SSRT values. As can be seen from Table 1, omissions to go stimuli were rare, and due to the implemented tracking algorithm, the mean inhibition rate approximated 50%. There were no significant outliers (defined as values that exceed three standard deviations from the mean) for any of the variables. The mean MAAS score of the final sample (*n* = 75) was 3.72 (range 1.6–5.6; SD = 0.74).

Assumptions for the planned analyses were met. Specifically, there was no substantial kurtosis or skewness regarding any of the variables (all within the −1–+1 range), and variances across conditions did not differ significantly (Mauchly’s test of sphericity was not significant for the within the subjects-factor condition).

There was a main effect of condition regarding SSRT, F(2,73) = 4.53, *p* = 0.014, ηp2 = 0.11. Post-hoc analyses (Bonferroni adjusted) showed that SSRT did not differ between the food condition and neutral condition (*p* = 0.489), or between the money condition and food condition (*p* = 0.554). SSRT was significantly higher in the money condition relative to the neutral condition (*p* = 0.012).

The MAAS score was not significantly correlated with SSRT in the neutral condition (Figure 1) and the food condition (Figure 2), respectively r(75) = −0.09, *p* = 0.470, 95% CI (−0.31, 0.15); r(75) = −0.07, *p* = 0.542, 95% CI (−0.29, 0.16). However, as depicted in Figure 3, MAAS score was negatively correlated with SSRT in the money condition, r(75) = −0.39, *p* = 0.001, 95% CI (−0.56, −0.18).

## 4. Discussion

Previous studies suggested a positive relationship between dispositional mindfulness and inhibitory control. However, it was not clear whether mindfulness is specifically associated with inhibitory control and whether reward context moderates this relationship. We found that the MAAS score was inversely related to the stop signal reaction time in the money condition, indicating a positive association between mindfulness and inhibitory control when perceiving money-related stimuli. Our results did not support a relationship between MAAS score and inhibitory control in a neutral context or in a different intrinsic reward context of palatable food.

Perhaps surprisingly, we did not observe a relationship between MAAS score and SSRT in the neutral context. At first glance, this might be perceived as incongruent with previous reports suggesting improved inhibitory control following mindfulness training. On the other hand, the effects of dispositional mindfulness might differ from mindfulness training. It should also be noted that studies that focused on the relationship between mindfulness training and inhibition-related measures yielded mixed results, some reported association with performance [5], while others did not [13,14]. Part of these inconsistent findings may be the variety of methodologies and implementations employed to assess inhibitory control. For instance, in a previous report, a go/no-go task with a 1:1 ratio of go trials relative to no-go trials was employed [13]. However, this may not have been optimal to trigger inhibitory control and identify an effect of mindfulness on the behavioral index of inhibitory performance [12]. And in Baily et al. [14], no mindfulness intervention was employed, but meditators were compared to non-meditators, making it difficult to disentangle specific from non-specific effects. Most studies that have rendered an effect have employed measures (e.g., response speed cost due to incongruence in stop task and failed to inhibit rate in response inhibition task) that are also affected by other processes, most notably attention.

Interestingly, and in contrast to our hypothesis, we found a significant moderate negative relationship between the MAAS score and SSRT in the money condition. This implies that higher dispositional mindfulness is associated with better inhibitory control after seeing money. Based on previous research utilizing the go/no-go task, including our own recent study, we expected the opposite. It is possible that the incongruent findings may be attributable to different inhibitory demands induced by the different tasks. To elaborate, in the reward condition of the GNG task, participants are required to inhibit depending on the color of the border of the reward-related stimulus. However, in the reward condition of our SST, participants are required to inhibit in response to a neutral stop stimulus presented subsequent to a reward-related go stimulus. In other words, in the GNG task, participants need to inhibit while they perceive a reward-related stimulus, whereas in the SST participants need to inhibit after perceiving a reward-related stimulus. It seems that with higher mindfulness, inhibitory control is challenged while perceiving a reward-related stimulus but facilitated subsequent to perceiving a reward-related stimulus. It should be emphasized that this effect was restricted to the money condition and was not mirrored in the food condition. It is not completely clear what explains this discrepancy. One potential possibility is that the difference is due to the type of reward. While palatable food can be regarded as an intrinsic reward, money is a learned reward (the reinforcing value is learned over time). On the other hand, it may be argued that money is universally rewarding, whereas the reward value of food-associated stimuli depends more on subject states. For instance, for some individuals, palatable food may be regarded as a reward, whereas others (e.g., when on a diet) may not regard it as a reward. This may also explain the higher variability in the palatable food context. Overall, our results may imply that the relationship between mindfulness and inhibitory control is restricted to a learned-reward context, though appropriate nuance should be applied when generalizing results.

One of the strengths of the current study is the employment of the stop signal paradigm, which effectively controls for differences in response speed, which is especially important when evaluating inhibitory control in a context of rewards where attentional bias plausibly plays a role. In addition, we used a counterbalanced within-subjects design, effectively controlling for potential order effects and individual differences. Lastly, one advantage was the online format and, as such, increased the ecological validity of the study.

Several potential limitations should be mentioned. First, it might be assumed that the online experiment platform is not as accurate in measuring reaction times as a controlled lab environment. Recent studies refute this claim and show that online platforms can be as valid and reliable as ordinary labs [28,36]. Still, potential distracting factors could have induced random error and increased response variability. However, inflated variability would only increase false-negative findings through decreased statistical power. We also excluded participants with invalid SSRTs keeping only participants that paid attention to the task [33]. The remaining sample showed a low omission rate to go stimuli and a high proportion of correct responses. In addition, the adaptive tracking algorithm was set up to ensure an approximate 50% correct inhibition rate which adds to the reliability of the estimated SSRT [31]. It should be emphasized that our results from the standard neutral condition largely overlap with our previous results obtained in a controlled lab environment [37]. To elaborate, response time variability is comparable between the offline (lab) and current online implementation, as well as obtained average SSRTs. The later showing a maximum 30 ms deviation with slightly shorter SSRTs in the online implementation. Mean response time is approximately 120 ms higher in the current online implementation. Indeed, the latter difference might be interpreted as indicating a potential slowing of responses in the current implementation, which could pose a threat to the validity of estimated SSRT. However, it is highly implausible that participants’ responses slowed as a function of time. Participants were explicitly instructed to respond as fast and accurate as possible to go-stimuli. In addition, if participants would slow down, then even with the tracking algorithm in place, the inhibit rate would logically exceed 50% which is not evident from our observed data.

A second limitation may be that our conditions differed not only with respect to reward type but also in terms of stimulus complexity. However, this notion is not supported by the data. To elaborate, it is known that increased stimulus/task complexity is associated with increased response times (as more complex stimuli and tasks require more processing time) [38]. However, we did not observe a significant response time difference between the two reward conditions.

Lastly, another potential limitation could be the use of the screening instrument for English language proficiency to exclude participants with below B1 English language proficiency. Logically, the instrument lacks some sensitivity due to the limited response options, and it relies on self-report in English. If this would have impacted results due to erroneous inclusion of participants with below B1 proficiency, it would most plausibly have resulted in higher randomness and variability in responses and associated lower statistical power. In that vein, if the use of the instrument would have affected results, the actual effects may plausibly be stronger, not weaker.

## 5. Conclusions

Taken together, results imply that when interpreting the relationship between mindfulness and inhibitory control, reward context should be taken into account. Future studies should ascertain whether the same applies to mindfulness training.

## Figures and Tables

**Figure 1 ijerph-19-02201-f001:**
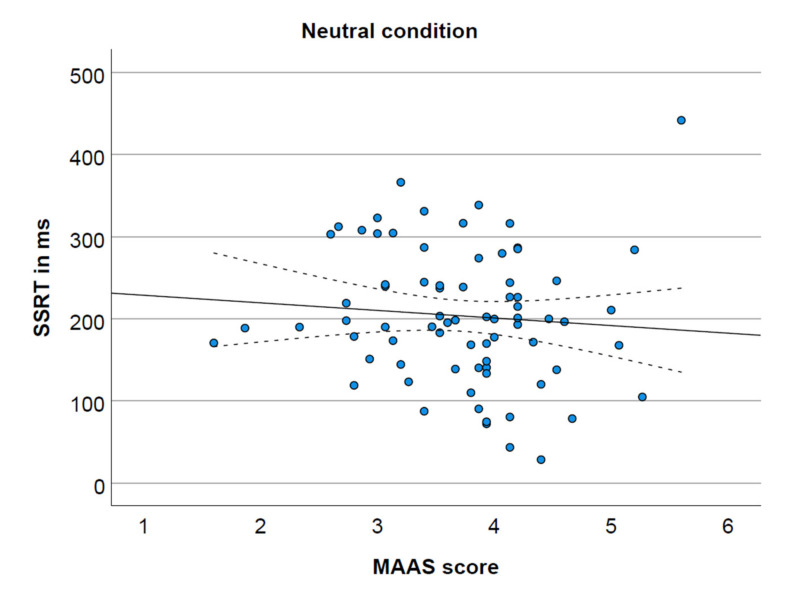
MAAS score and SSRT in the neutral condition (non-significant), including 95% confidence interval.

**Figure 2 ijerph-19-02201-f002:**
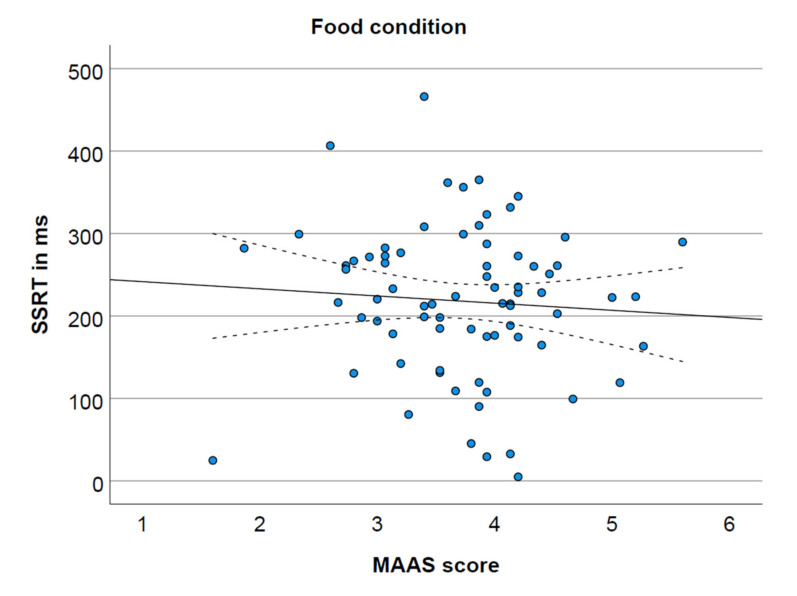
MAAS score and SSRT in the food condition (non-significant), including 95% confidence interval.

**Figure 3 ijerph-19-02201-f003:**
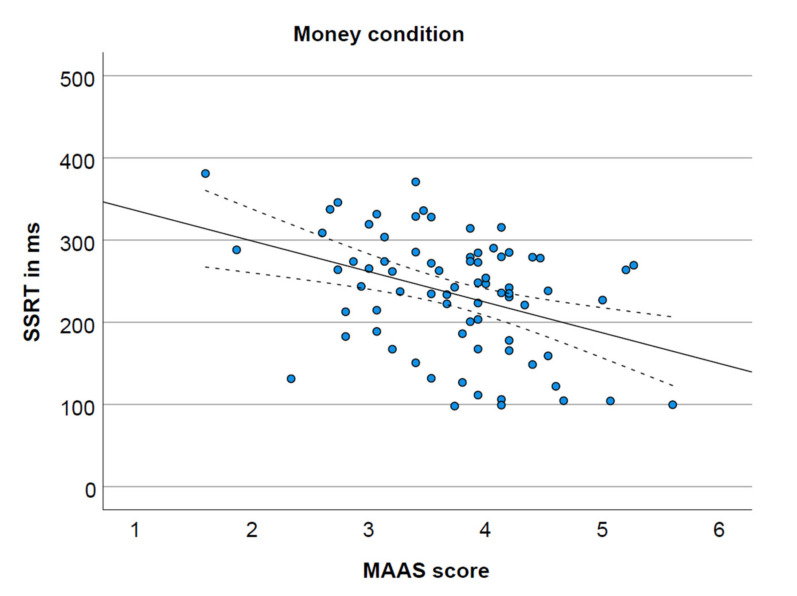
Significant negative correlation between MAAS score and SSRT in the money condition (*p* = 0.004), including 95% confidence interval.

**Table 1 ijerph-19-02201-t001:** Average stop signal task performance in three conditions.

	Neutral	Food	Money
SSRT (ms)	204 (81)	218 (90)	235 (72)
Mean RT (ms)	795 (154)	804 (153)	781 (156)
SD RT (ms)	171 (54)	187 (60)	174 (53)
Correct Inhibition	52% (11%)	51% (11%)	49% (11%)
Correct Answer	94% (11%)	86% (16%)	89% (12%)
Omission	4% (6%)	6% (9%)	5% (7%)

Note: Numbers represent means (standard deviations); *n* = 75; SSRT: Stop Signal Reaction Time; RT: Reaction Time; SD RT: average of individuals’ response time variability indexed by the standard deviation of RT to go stimuli; Correct inhibition: percent inhibitions in stop trials corrected for omissions.

## Data Availability

All materials and data are available via the Open Science Framework (www.osf.io), with identifier: doi:10.17605/OSF.IO/Z8DPF.

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
