# Peer review of "Dispositional Mindfulness and Inhibitory Control after Perceiving Neutral, Food and Money Related Stimuli"

_ijerph, 2022, doi:10.3390/ijerph19042201_

Round 1

Reviewer 1 Report

Review of IJERPH-1581696 "Dispositional Mindfulness and inhibitory control after perceiving neutral, food and money related stimuli "

The present manuscript describes a study which examined the association between a measure of trait mindfulness and inhibitory control, which utilized an online sample. Specifically, this research used a Stop Signal Task (SST) to test how mindfulness was associated with inhibitory control when presented with imagery of a monetary or food reward. Results of this study were that mindfulness was associated with more inhibitory control in the monetary but not food conditions. The authors posit that this means that mindfulness may impact inhibitory control in different reward contexts. The manuscript is pithy, and the authors present their results clearly. Furthermore, this research is fairly novel and has implications for future research. However, some aspects of the sample need to be elucidated and the conclusions drawn are but beyond the scope of the presented data. What follows are a list of critiques of the manuscript:     

Major Critiques

Methods

  1. More information on the sample is needed. Based on what is described, this sample was derived from English-proficient people across the globe. That is acceptable, but it would be useful to know more about the sample as it may impact the results. For example, whether English was participants’ first language.
  2. The current English proficiency screening measure seems to lack sensitivity. Was there any other means of determining participants’ English proficiency? If not, this seems like it should be discussed as a possible limitation.

Conclusion

  1. The authors state “Taken together, results imply that mindfulness training should take into account the role of reward context as the relationship between mindfulness and inhibitory control seems to depend on reward context.” I don’t really think the authors can make this claim given their data. The current research did not manipulate mindfulness- it merely measured “dispositional” mindfulness. At most they could say that mindfulness may be associated with inhibitory control depending on context. More experimental research is needed on this topic.

Minor Critiques

Introduction

  1. Please define “dispositional mindfulness.”
  2. There are times throughout the manuscript where either the term “ability to stop” or “inhibitory [control]” are used– please be consistent in language.
  3. The acronym for Go/no-go is defined towards the end of the introduction after first using term go/no go. It may be easier to define acronym at first use of the term and use the acronym thereafter.
  4. The wording of the second hypothesis is confusing to me. Do the authors mean that they expect to see an increase in reaction time with increases in mindfulness?

Methods

  1. What was the minimum sample size calculated to reach statistical power? Was this sample size reached?
  2. What is “palatable food”? Given this is a (presumably) an international sample how can you tell if participants perceived the food as palatable? Was there any measure of this?
  3. Similarly what currency was used? I would imagine people for whom this is not the currency they used in their country would not have a “learned response” to this type of money.

Author Response

The present manuscript describes a study which examined the association between a measure of trait mindfulness and inhibitory control, which utilized an online sample. Specifically, this research used a Stop Signal Task (SST) to test how mindfulness was associated with inhibitory control when presented with imagery of a monetary or food reward. Results of this study were that mindfulness was associated with more inhibitory control in the monetary but not food conditions. The authors posit that this means that mindfulness may impact inhibitory control in different reward contexts. The manuscript is pithy, and the authors present their results clearly. Furthermore, this research is fairly novel and has implications for future research. However, some aspects of the sample need to be elucidated and the conclusions drawn are but beyond the scope of the presented data. What follows are a list of critiques of the manuscript:     

Major Critiques

Methods

  1. More information on the sample is needed. Based on what is described, this sample was derived from English-proficient people across the globe. That is acceptable, but it would be useful to know more about the sample as it may impact the results. For example, whether English was participants’ first language.

> Thank you for raising this important point. We now made clear that most participants were native English Speakers. Only those participants with sufficient proficiency were allowed to participate. In the “participants” section we now added the following: “In total 64 participants were from the United Kingdom, and were native English speakers. The other 23 participants were from elsewhere, and not necessarily native English speakers. However, participants were excluded from the subsequent analyses if they did not meet at least a B1 independent level of lingual proficiency in English (according to the Language proficiency questionnaire).”

  1. The current English proficiency screening measure seems to lack sensitivity. Was there any other means of determining participants’ English proficiency? If not, this seems like it should be discussed as a possible limitation.

> Thank you for this comment. Indeed, the proficiency screening instrument provides a relative crude estimate of general English language proficiency. However, our primary aim concerned excluding those participants that did not have a minimal independent level of English proficiency. Certainly, a more objective measure could have been more optimal, but then the assessment time would increase plausibly resulting in a higher attrition rate. Nevertheless we indicated the use of the self-report measure of lingual proficiency as a potential limitation in the discussion. Specifically, we mention in the discussion section:  “Lastly, another potential limitation could be the use of the screening instrument for English language proficiency to exclude participants with below B1 English language proficiency. Logically, the instrument lacks some sensitivity due to the limited response options, and it relies on self-report in English. If this would have impacted results due to erroneous inclusion of participants with below B1 proficiency, it would most plausibly have resulted in higher randomness and variability in responses and associated lower statistical power. In that vein, if the use of the instrument would have affected results, the actual effects may plausibly be stronger not weaker.”

Conclusion

  1. The authors state “Taken together, results imply that mindfulness training should take into account the role of reward context as the relationship between mindfulness and inhibitory control seems to depend on reward context.” I don’t really think the authors can make this claim given their data. The current research did not manipulate mindfulness- it merely measured “dispositional” mindfulness. At most they could say that mindfulness may be associated with inhibitory control depending on context. More experimental research is needed on this topic.

 > Thank you for this observation. We agree, and have adjusted the conclusion to better fit the observed results. We state: “Taken together, results imply that when interpreting the relationship between mindfulness and inhibitory control, reward context should be taken into account. Future studies should ascertain whether the same applies to mindfulness training.”

Minor Critiques

Introduction

  1. Please define “dispositional mindfulness.”

> Indeed, noting the broad audience of the journal, not everyone may be familiar with the term “dispositional”, hence in addition to the definition of mindfulness, we now also stated that dispositional refers to trait. Specifically, we state at the first instance of “dispositional” the following: “Less is known about dispositional, or in other words, trait mindfulness in relation to inhibitory control,”

  1. There are times throughout the manuscript where either the term “ability to stop” or “inhibitory [control]” are used– please be consistent in language.

> Thank you for noticing, we increased the consistency.

  1. The acronym for Go/no-go is defined towards the end of the introduction after first using term go/no go. It may be easier to define acronym at first use of the term and use the acronym thereafter.

> Thank you for this observation, we have made the suggested change.

  1. The wording of the second hypothesis is confusing to me. Do the authors mean that they expect to see an increase in reaction time with increases in mindfulness?

 > Indeed, the formulation was a bit ambiguous. We have increased the specificity of the second hypothesis and formulated it as follows: “Secondly, we hypothesized that higher mindfulness would be associated with challenged inhibitory control in both reward conditions, reflected by a positive correlation between trait mindfulness and stop signal reaction time in these conditions.”

Methods

  1. What was the minimum sample size calculated to reach statistical power? Was this sample size reached?

> Thank you for this important comment. We stated that we performed an apriori sample size estimation, however, we understand that more details would be needed. We aimed to detect moderate effect sizes, and the final sample size after exclusion of participants with erroneous data was sufficient to that aim. We made this clearer in the participants section: “The target sample size was estimated with G*Power [25]. Specifically, with alpha set at 0.05 and power at 0.8, the initial target sample size was 80 participants. In total, 87 participants completed the experiments, and were provided with a 2.50 GBP monetary compensation for participating. Participants were excluded from the subsequent analyses if they did not meet at least a B1 independent level of lingual proficiency in English (according to the Language proficiency questionnaire). In addition, we excluded participants with an invalid estimate of the SSRT (see the materials section for details). The final sample consisted of 75 participants (31 men, 44 women) between 18-50 years old (M=30, SD=9). This sample size was sufficient to detect moderate correlations of r>0.32.”

  1. What is “palatable food”? Given this is a (presumably) an international sample how can you tell if participants perceived the food as palatable? Was there any measure of this?

> Thank you for this valuable comment. The definition of palatable food tends to vary, and we specified in the introduction that palatable foods are those foods high in sugar and fat content. Indeed, what is perceived as “palatable” may vary as a function of culture. In that vein, we should note that we chose those stimuli which are generally familiar to most individuals across cultures. We mentioned in the introduction the following: “In the reward conditions, stimuli were related to generally recognizable palatable foods, foods that are high in fat and sugar (food condition),[…]”, and in the Materials section:  “Similar to a previous implementation [4], in the food condition the go stimulus was one of four equiprobable pictures of palatable food (chips, chocolate-chip cookies, nuts, chocolate),[…]”

  1. Similarly what currency was used? I would imagine people for whom this is not the currency they used in their country would not have a “learned response” to this type of money.

> Thank you for this related point. The instruction to participants clearly indicated that they would be presented with pictures of money in the money condition, and pictures were recognizable as representing money (bills, coins), and these were presented in currencies that most individuals are generally familiar with (Dollars/Euros). We now specifically added in the materials section that: “In the money condition, the go stimulus was one of four equiprobable pictures of money most individuals are familiar with (Euros and Dollars),[…]”.

Reviewer 2 Report

The study design of this research is well designed to overcome previous limitations related to mindfulness and inhibitory control. For this reason, the study notably advances the science related to the correlation of outcome measures with mindfulness scores by MAAS. 

Please consider improving your Figures or Figure Legends for Figures 1–3 by including the statistical results with the graphs.

You appear to have used a Fisher's ANOVA for data analysis. Use of this ANOVA requires two assumptions confirmed by statistical testing. The first is a normal distribution of all data included in the analysis. This should be determined using a Shapiro–Wilk test or similar and the results reported in general to confirm that no datasets significantly diverged from a normal distribution curve. If a normal distribution is not present, the a Kruskal–Wallis ANOVA is indicated. Next, the datasets must have homogeneity of variance as determined by a Levene test. If your datasets do not have equal variance, then the Fisher's ANOVA will report a false positive p-value which can skew your results and interpretations. Please report the general results of the Levene test to assure the reader that your p-values can be strictly interpreted. If the Levene test is not satisfied, then a Welch's ANOVA is indicated. Finally, please state the type of post-hoc testing used to correct for the multiple comparisons performed. 

While not necessary, consider describing the Cohen's d effect size for comparisons that are not statistically significant. This can be used by the reader to estimate sample sizes required to reach significance based on the obtained results, providing the reader with an important conclusion and allowing your study to be used as the basis for future study designs.

Your Discussion provides good context. Your limitations are well described, and your Conclusion is appropriate.

Author Response

The study design of this research is well designed to overcome previous limitations related to mindfulness and inhibitory control. For this reason, the study notably advances the science related to the correlation of outcome measures with mindfulness scores by MAAS. 

Please consider improving your Figures or Figure Legends for Figures 1–3 by including the statistical results with the graphs.

> Thank you for this comment, we have added information regarding the significance of the observed relationship in the figure title.

You appear to have used a Fisher's ANOVA for data analysis. Use of this ANOVA requires two assumptions confirmed by statistical testing. The first is a normal distribution of all data included in the analysis. This should be determined using a Shapiro–Wilk test or similar and the results reported in general to confirm that no datasets significantly diverged from a normal distribution curve. If a normal distribution is not present, the a Kruskal–Wallis ANOVA is indicated. Next, the datasets must have homogeneity of variance as determined by a Levene test. If your datasets do not have equal variance, then the Fisher's ANOVA will report a false positive p-value which can skew your results and interpretations. Please report the general results of the Levene test to assure the reader that your p-values can be strictly interpreted. If the Levene test is not satisfied, then a Welch's ANOVA is indicated.

> Thank you for these important suggestions pertaining to the statistical approaches. Indeed, prior to performing the parametric tests, we first ascertained whether the key assumptions were met, however an explicit statement was omitted. We now added more details. We state: “Specifically, there was no substantial kurtosis or skewness regarding any of the variables (all within the -1 - +1 range), and variances across condition did not differ significantly (Mauchly’s test of sphericity was not significant for the within subjects-factor condition).”.

Finally, please state the type of post-hoc testing used to correct for the multiple comparisons performed. 

> Thank you for this comment. Indeed from a strict statistical perspective, correction for multiple pairwise comparisons would have been better. Initially we performed the LSD post-hoc pairwise comparisons which may be too liberal (though the number of tests were very limited). Subsequent to your suggestion, we have now corrected for the multiple pairwise comparisons using Bonferroni correction. We state in the results: “Post-hoc analyses (Bonferroni adjusted) showed that SSRT did not differ between the food condition and neutral condition (p=0.489), or between the money condition and food condition (p=0.554). SSRT was significantly higher in the money condition relative to the neutral condition (p=0.012).

While not necessary, consider describing the Cohen's d effect size for comparisons that are not statistically significant. This can be used by the reader to estimate sample sizes required to reach significance based on the obtained results, providing the reader with an important conclusion and allowing your study to be used as the basis for future study designs.

> Thank you for this suggestion. We have provided effect size measures relevant for repeated measures ANOVA’s where applicable (partial eta squared), to avoid redundancy we omitted the Cohen’s d measure, though these can be calculated from the reported means and standard deviations across condition.

Your Discussion provides good context. Your limitations are well described, and your Conclusion is appropriate.

Reviewer 3 Report

Define "stroop task" for those who are beyond your field. 

Be consistent: switch to GNG earlier, if you're going to do it at all. 

Overall, it seems like there's some value to the method that you used and the question that you asked, but it would seem to be difficult to make any sort of conclusion about the question based on the subjective regard of the reward (discussed in the penultimate section). Because the question of "reward" is so subjective, especially relative to an online presentation of something, the more interesting conjecture concerning intrinsic and learned rewards seem quite difficult to accurately measure given the constraints of the above. 

Also: if you revise this, writing your results in smaller paragraphs would allow readers to more usefully process the information than the long blocks currently included. 

Author Response

Define "stroop task" for those who are beyond your field. 

> Thank you for this comment, indeed this should have been explained. However, subsequent to a comment of another reviewer, we now chose to omit reference to the stroop task, and more specifically focus on the go/no-go task literature.

Be consistent: switch to GNG earlier, if you're going to do it at all. 

> Thank you for this observation, we added more focus on the GNG task, and provided a more detailed explanation of the GNG task early in the introduction. Specifically, we now state: “Specifically, mindfulness training has been shown to enhance go/no-go type task performance [5], which may suggest inhibitory improvement. In such task, visual go stimuli to which a simple response is required are presented sequentially. Infrequently, a visual no-go stimulus is presented to which a response has to be withhold”

Overall, it seems like there's some value to the method that you used and the question that you asked, but it would seem to be difficult to make any sort of conclusion about the question based on the subjective regard of the reward (discussed in the penultimate section). Because the question of "reward" is so subjective, especially relative to an online presentation of something, the more interesting conjecture concerning intrinsic and learned rewards seem quite difficult to accurately measure given the constraints of the above. 

> Thank you for raising this important issue. We should note, with reference to our introduction, that the types of stimuli we have employed are known to generally trigger brain circuitry important for reward processing. But indeed, as also raised by another reviewer, perceived reward value of stimuli may vary as a function of culture. We took this into account, and employed those stimuli that are common, especially given the majority of the target sample (mostly individuals from the UK). We now provided more information regarding the sample, and made clear in the materials section what exact stimuli we used for the reward conditions.

Also: if you revise this, writing your results in smaller paragraphs would allow readers to more usefully process the information than the long blocks currently included. 

> Thank you for this comment. We have now improved the readability of the results section by subdividing the different elements in smaller paragraphs.